# Predicting User Activity Level In Point Processes With Mass Transport Equation

**Yichen Wang$^\diamond$, Xiaojing Ye$^*$, Hongyuan Zha$^\diamond$, Le Song$^{\diamond\dagger}$**
$^\diamond$College of Computing, Georgia Institute of Technology
$^*$ School of Mathematics, Georgia State University
$^\dagger$ Ant Financial
{yichen.wang}@gatech.edu, xye@gsu.edu
{zha,lsong}@cc.gatech.edu

## Abstract

Point processes are powerful tools to model user activities and have a plethora of applications in social sciences. Predicting user activities based on point processes is a central problem. However, existing works are mostly problem specific, use heuristics, or simplify the stochastic nature of point processes. In this paper, we propose a framework that provides an efficient estimator of the probability mass function of point processes. In particular, we design a key reformulation of the prediction problem, and further derive a differential-difference equation to compute a conditional probability mass function. Our framework is applicable to general point processes and prediction tasks, and achieves superb predictive and efficiency performance in diverse real-world applications compared to the state of the art.

## 1 Introduction

Online social platforms, such as Facebook and Twitter, enable users to post opinions, share information, and influence peers. Recently, user-generated event data archived in fine-grained temporal resolutions are becoming increasingly available, which calls for expressive models and algorithms to understand, predict and distill knowledge from complex dynamics of these data. Particularly, temporal point processes are well-suited to model the event pattern of user behaviors and have been successfully applied in modeling event sequence data [6, 10, 12, 21, 23, 24, 25, 26, 27, 28, 33].

A fundamental task in social networks is to predict user activity levels based on learned point process models. Mathematically, the goal is to compute $\mathbb{E}[f(N(t))]$, where $N(\cdot)$ is a given point process that is learned from user behaviors, $t$ is a fixed future time, and $f$ is an application-dependent function. A framework for doing this is critically important. For example, for social networking services, an accurate inference of the number of reshares of a post enables the network moderator to detect trending posts and improve its content delivery networks [13, 32]; an accurate estimate of the change of network topology (the number of new followers of a user) facilitates the moderator to identify influential users and suppress the spread of terrorist propaganda and cyber-attacks [12]; an accurate inference of the activity level (number of posts in the network) allows us to gain fundamental insight into the predictability of collective behaviors [22]. Moreover, for online merchants such as Amazon, an accurate estimate of the number of future purchases of a product helps optimizing future advertisement placements [10, 25].

Despite the prevalence of prediction problems, an accurate prediction is very challenging for two reasons. First, the function $f$ is arbitrary. For instance, to evaluate the homogeneity of user activities, we set $f(x) = x\log(x)$ to compute the Shannon entropy; to measure the distance between a predicted activity level and a target $x^*$, we set $f(x) = (x - x^*)^2$. However, most works [8, 9, 13, 30, 31, 32] are problem specific and only designed for the simple task with $f(x) = x$; hence these works are

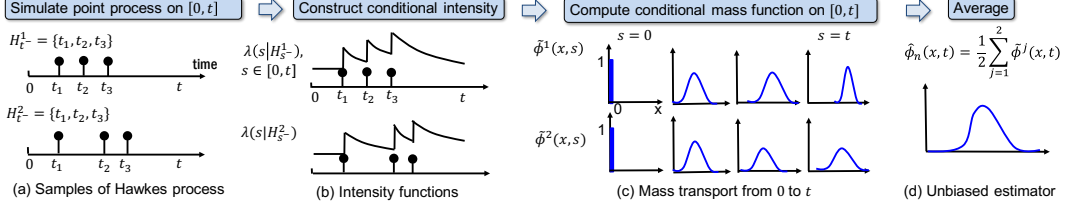

(a) Samples of Hawkes process     (b) Intensity functions     (c) Mass transport from 0 to $t$     (d) Unbiased estimator

Figure 1: An illustration of HYBRID using Hawkes process (Eq. 1). Our method first generates two samples $\{\mathcal{H}_{t-}^i\}$ of events; then it constructs intensity functions; with these inputs, it computes conditional probability mass functions $\tilde{\phi}^i(x,s) := \mathbb{P}[N(s) = x|\mathcal{H}_{s-}^i]$ using a mass transport equation. Panel (c) shows the transport of conditional mass at four different times (the initial probability mass $\tilde{\phi}(x,0)$ is an indicator function $\mathbb{I}[x = 0]$, as there is no event with probability one). Finally, the average of conditional mass functions yields our estimator of the probability mass.

not generalizable. Second, point process models typically have intertwined stochasticity and can co-evolve over time [12, 25], *e.g.*, in the influence propagation problem, the information diffusion over networks can change the structure of networks, which adversely influences the diffusion process [12]. However, previous works often ignore parts of the stochasticity in the intensity function [29] or make heuristic approximations [13, 32]. Hence, there is an urgent need for a method that is applicable to an arbitrary function $f$ and keeps all the stochasticity in the process, which is largely nonexistent to date.

We propose HYBRID, a generic framework that provides an efficient estimator of the probability mass of point processes. Figure 1 illustrates our framework. We also make the following contributions:

- **Unifying framework**. Our framework is applicable to general point processes and does not depend on specific parameterization of intensity functions. It incorporates all stochasticity in point processes and is applicable to prediction tasks with an arbitrary function $f$.

- **Technical challenges**. We reformulate the prediction problem and design a random variable with reduced variance. To derive an analytical form of this random variable, we also propose a mass transport equation to compute the conditional probability mass of point processes. We further transform this equation to an Ordinary Differential Equation and provide a scalable algorithm.

- **Superior performance**. Our framework significantly reduces the sample size to estimate the probability mass function of point processes in real-world applications. For example, to infer the number of tweeting and retweeting events of users in the co-evolution model of information diffusion and social link creation [12], our method needs $10^3$ samples and 14.4 minutes, while Monte Carlo needs $10^6$ samples and 27.8 hours to achieve the same relative error of 0.1.

## 2 Background and preliminaries

**Point processes**. A temporal point process [1] is a random process whose realization consists of a set of discrete events $\{t_k\}$, localized in time. It has been successfully applied to model user behaviors in social networks [16, 17, 19, 23, 24, 25, 28, 30]. It can be equivalently represented as a counting process $N(t)$, which records the number of events on $[0, t]$. The counting process is a right continuous step function, *i.e.*, if an event happens at $t$, $N(t) - N(t^-) = 1$.

Let $\mathcal{H}_{t-} = \{t_k|t_k < t\}$ be the history of events happened up to time $t$. An important way to characterize point processes is via the conditional intensity function $\lambda(t) := \lambda(t|\mathcal{H}_{t-})$, a stochastic model for the time of the next event given the history. Formally, $\lambda(t)$ is the conditional probability of observing an event in $[t, t + \mathrm{d}t)$ given events on $[0, t)$, *i.e.*, $\mathbb{P}\{\text{event in } [t, t + \mathrm{d}t)|\mathcal{H}_{t-}\} = \mathbb{E}[\mathrm{d}N(t)|\mathcal{H}_{t-}] := \lambda(t)\mathrm{d}t$, where $\mathrm{d}N(t) \in \{0, 1\}$.

The intensity function is designed to capture the phenomena of interest. Some useful forms include (i) Poisson process: the intensity is a deterministic function, and (ii) Hawkes process [15]: it captures the mutual excitation phenomena between events and its intensity is parameterized as

$$\lambda(t) = \eta + \alpha \sum\nolimits_{t_k \in \mathcal{H}_{t-}} \kappa(t - t_k), \tag{1}$$

where $\eta \geqslant 0$ is the baseline intensity; the trigging kernel $\kappa(t) = \exp(-t)$ models the decay of past events' influence over time; $\alpha \geqslant 0$ quantifies the strength of influence from each past event. Here, the occurrence of each historical event increases the intensity by a certain amount determined by $\kappa(t)$ and $\alpha$, making $\lambda(t)$ history-dependent and a stochastic process by itself.

**Monte Carlo (MC).** To compute the probability mass of a point process, MC simulates $n$ realizations of history $\{\mathcal{H}_t^i\}$ using the thinning algorithm [20]. The number of events in sample $i$ is defined as $N^i(t) = |\mathcal{H}_t^i|$. Let $\phi(x, t) := \mathbb{P}[N(t) = x]$, where $x \in \mathbb{N}$, be the probability mass. Then its estimator $\hat{\phi}_n^{mc}(x, t)$ and the estimator $\hat{\mu}_n^{mc}(t)$ for $\mu(t) := \mathbb{E}[f(N(t))]$ are defined as $\hat{\phi}_n^{mc}(x, t) = \frac{1}{n} \sum_i \mathbb{I}[N^i(t) = x]$ and $\hat{\mu}_n^{mc}(t) = \frac{1}{n} \sum_i f(N^i(t))$. The root mean square error (RMSE) is defined as

$$\varepsilon(\hat{\mu}_n^{mc}(t)) = \sqrt{\mathbb{E}[\hat{\mu}_n^{mc}(t) - \mu(t)]^2} = \sqrt{\mathbb{VAR}[f(N(t))]/n}. \tag{2}$$

# 3 Solution overview

Given an arbitrary point process $N(t)$ that is learned from data, existing prediction methods for computing $\mathbb{E}[f(N(t))]$ have three major limitations:

- **Generalizability**. Most methods [8, 9, 13, 30, 31, 32] only predict $\mathbb{E}[N(t)]$ and are not generalizable to an arbitrary function $f$. Moreover, they typically rely on specific parameterizations of the intensity functions, such as the reinforced Poisson process [13] and Hawkes process [5, 32]; hence they are not applicable to general point processes.

- **Approximation and heuristics**. These works also ignore parts of the stochasticity in the intensity functions [29] or make heuristic approximations to the point process [13, 32]. Hence the accuracy is limited by the approximations and heuristic corrections.

- **Large sample size**. The MC method overcomes the above limitations since it has an unbiased estimator of the probability mass. However, the high stochasticity in point processes leads to a large value of $\mathbb{VAR}[f(N(t))]$, which requires a large number of samples to achieve a small error.

To address these challenges, we propose a generic framework with a novel estimator of the probability mass, which has a smaller sample size than MC. Our framework has the following key steps.

**I**. **New random variable**. We design a random variable $g(\mathcal{H}_{t^-})$, a conditional expectation given the history. Its variance is guaranteed to be smaller than that of $f(N(t))$. For a fixed number of samples, the error of MC is decided by the variance of the random variable of interest, as shown in (2). Hence, to achieve the same error, applying MC to estimate the new objective $\mathbb{E}_{\mathcal{H}_{t^-}}[g(\mathcal{H}_{t^-})]$ requires smaller number of samples compared with the procedure that directly estimates $\mathbb{E}[f(N(t))]$.

**II**. **Mass transport equation**. To compute $g(\mathcal{H}_{t^-})$, we derive a differential-difference equation that describes the evolutionary dynamics of the conditional probability mass $\mathbb{P}[N(t) = x | \mathcal{H}_{t^-}]$. We further formulate this equation as an Ordinary Differential Equation, and provide a scalable algorithm.

# 4 Hybrid inference machine with probability mass transport

In this section, we present technical details of our framework. We first design a new random variable for prediction; then we propose a mass transport equation to compute this random variable analytically. Finally, we combine the mass transport equation with the sampling scheme to compute the probability mass function of general point processes and solve prediction tasks with an arbitrary function $f$.

## 4.1 New random variable with reduced variance

We reformulate the problem and design a new random variable $g(\mathcal{H}_{t^-})$, which has a smaller variance than $f(N(t))$ and the same expectation. To do this, we express $\mathbb{E}[f(N(t))]$ as an iterated expectation

$$\mathbb{E}[f(N(t))] = \mathbb{E}_{\mathcal{H}_{t^-}} \left[ \mathbb{E}_{N(t)|\mathcal{H}_{t^-}} \left[ f(N(t)) | \mathcal{H}_{t^-} \right] \right] = \mathbb{E}_{\mathcal{H}_{t^-}} \left[ g(\mathcal{H}_{t^-}) \right], \tag{3}$$

where $\mathbb{E}_{\mathcal{H}_{t^-}}$ is w.r.t. the randomness of the history and $\mathbb{E}_{N(t)|\mathcal{H}_{t^-}}$ is w.r.t. the randomness of the point process given the history. We design the random variable as a conditional expectation given the history: $g(\mathcal{H}_{t^-}) = \mathbb{E}_{N(t)|\mathcal{H}_{t^-}}[f(N(t))|\mathcal{H}_{t^-}]$. Theorem 1 shows that it has a smaller variance.

**Theorem 1.** *For time $t > 0$ and an arbitrary function $f$, we have $\mathbb{VAR}[g(\mathcal{H}_{t^-})] < \mathbb{VAR}[f(N(t))]$.*

Theorem 1 extends the Rao-Blackwell (RB) theorem [3] to point processes. RB says that if $\hat{\theta}$ is an estimator of a parameter $\theta$ and $T$ is a sufficient statistic for $\theta$; then $\mathbb{VAR}[\mathbb{E}[\hat{\theta}|T]] \leqslant \mathbb{VAR}[\hat{\theta}]$, *i.e.*, the sufficient statistic reduces uncertainty of $\hat{\theta}$. However, RB is not applicable to point processes since it studies a different problem (improving the estimator of a distribution's parameter), while we focus on the prediction problem for general point processes, which introduces two new technical challenges:

(i) Is there a definition in point processes whose role is similar to the sufficient statistic in RB? Our first contribution shows that the history $\mathcal{H}_{t^-}$ contains all the necessary information in a point process and reduces the uncertainty of $N(t)$. Hence, $g(\mathcal{H}_{t^-})$ is an improved variable for prediction. Moreover, in contrast to the RB theorem, the inequality in Theorem 1 is *strict* because the counting process $N(t)$ is right-continuous in time $t$ and not predictable [4] (a predictable process is measurable w.r.t. $\mathcal{H}_{t^-}$, such as the processes that are left-continuous). Appendix C contains details on the proof.

(ii) Is $g(\mathcal{H}_{t^-})$ computable for *general* point processes and an *arbitrary* function $f$? An efficient computation will enable us to estimate $\mathbb{E}_{\mathcal{H}_{t^-}}[g(\mathcal{H}_{t^-})]$ using the sampling method. Specifically, let $\hat{\mu}_n(t) = \frac{1}{n} \sum_i g(\mathcal{H}_{t^-}^i)$ be the estimator computed from $n$ samples; then from the definition of RMSE in (2), this estimator has smaller error than MC: $\varepsilon(\hat{\mu}_n(t)) < \varepsilon(\hat{\mu}_n^{mc}(t))$.

However, the challenge in our new formulation is that it seems very hard to compute this conditional expectation, as one typically needs another round of sampling, which is undesirable as it will increase the variance of the estimator. To address this challenge, next we propose a mass transport equation.

## 4.2 Transport equation for conditional probability mass function

We present a novel mass transport equation that computes the conditional probability mass $\tilde{\phi}(x,t) := \mathbb{P}[N(t) = x|\mathcal{H}_{t^-}]$ of general point processes. With this definition, we derive an analytical expression for the conditional expectation: $g(\mathcal{H}_{t^-}) = \sum_x f(x)\tilde{\phi}(x,t)$. The transport equation is as follows.

**Theorem 2** (Mass Transport Equation for Point Processes). *Let $\lambda(t) := \lambda(t|\mathcal{H}_{t^-})$ be the conditional intensity function of the point process $N(t)$ and $\tilde{\phi}(x,t) := \mathbb{P}[N(t) = x|\mathcal{H}_{t^-}]$ be its conditional probability mass function; then $\tilde{\phi}(x,t)$ satisfies the following differential-difference equation:*

$$\underbrace{\tilde{\phi}_t(x,t) := \frac{\partial \tilde{\phi}(x,t)}{\partial t}}_{\text{rate of change in conditional mass}} = \begin{cases} -\lambda(t)\tilde{\phi}(x,t) & \text{if } x = 0 \\ \underbrace{-\lambda(t)\tilde{\phi}(x,t)}_{\text{loss in mass, at rate } \lambda(t)} + \underbrace{\lambda(t)\tilde{\phi}(x-1,t)}_{\text{gain in mass, at rate } \lambda(t)} & \text{if } x = 1, 2, 3, \cdots \end{cases} \quad (4)$$

**Proof sketch.** For the simplicity of notation, we set the right-hand-side of (4) to be $\mathcal{F}[\tilde{\phi}]$, where $\mathcal{F}$ is a functional operator on $\tilde{\phi}$. We also define the inner product between functions $u : \mathbb{N} \to \mathbb{R}$ and $v : \mathbb{N} \to \mathbb{R}$ as $(u,v) := \sum_x u(x)v(x)$. The main idea in our proof is to show that the equality $(v, \tilde{\phi}_t) = (v, \mathcal{F}[\tilde{\phi}])$ holds for any test function $v$; then $\tilde{\phi}_t = \mathcal{F}[\tilde{\phi}]$ follows from the fundamental lemma of the calculus of variations [14]. Specifically, the proof contains two parts as follows.

We first prove $(v, \tilde{\phi}_t) = (\mathcal{B}[v], \tilde{\phi})$, where $\mathcal{B}[v]$ is a functional operator defined as $\mathcal{B}[v] = (v(x + 1) - v(x))\lambda(t)$. This equality can be proved by the property of point processes and the definition of conditional mass. Second, we show $(\mathcal{B}[v], \tilde{\phi}) = (v, \mathcal{F}[\tilde{\phi}])$ using a variable substitution technique. Mathematically, this equality means $\mathcal{B}$ and $\mathcal{F}$ are *adjoint* operators on the function space. Combining these two equalities yields the mass transport equation. Appendix A contains details on the proof.

**Mass transport dynamics**. This differential-difference equation describes the time evolution of the conditional mass. Specifically, the differential term $\tilde{\phi}_t$, *i.e.*, the instantaneous rate of change in the probability mass, is equal to a first order difference equation on the right-hand-side. This difference equation is a summation of two terms: (i) the negative loss of its own probability mass $\tilde{\phi}(x,t)$ at rate $\lambda(t)$, and (ii) the positive gain of probability mass $\tilde{\phi}(x-1,t)$ from last state $x-1$ at rate $\lambda(t)$. Moreover, since initially no event happens with probability one, we have $\tilde{\phi}(x,0) = \mathbb{I}[x = 0]$. Solving this transport equation on $[0,t]$ essentially transports the initial mass to the mass at time $t$.

| **Algorithm 1:** CONDITIONAL MASS FUNCTION | **Algorithm 2:** HYBRID MASS TRANSPORT |
|---|---|
| **Input**: $\mathcal{H}_{t^-} = \{t_k\}_{k=1}^K$, $\Delta\tau$, set $t = t_{K+1}$ | **Input**: Sample size $n$, time $t$, $\Delta\tau$ |
| **Output**: Conditional probability mass function $\tilde{\phi}(t)$ | **Output**: $\hat{\mu}_n(t), \hat{\phi}_n(x,t)$ |
| **for** $k = 0, \cdots K$ **do** | Generate $n$ samples of point process: $\{\mathcal{H}_{t^-}^i\}_{i=1}^n$; |
| $\quad$ Construct $\lambda(s)$ and $\boldsymbol{Q}(s)$ on $[t_k, t_{k+1}]$ ; | **for** $i = 1, \cdots, n$ **do** |
| $\quad$ $\tilde{\phi}(t_{k+1}) = \text{ODE45}[\tilde{\phi}(t_k), \boldsymbol{Q}(s), \Delta\tau)]$ (RK Alg); | $\quad$ $\tilde{\phi}^i(x,t) = \text{COND-MASS-FUNC}(\mathcal{H}_{t^-}^i, \Delta\tau)$; |
| **end** | **end** |
| Set $\tilde{\phi}(t) = \tilde{\phi}(t_{K+1})$ | $\hat{\phi}_n(x,t) = \frac{1}{n}\sum_i \tilde{\phi}^i(x,t)$, $\hat{\mu}_n(t) = \sum_x f(x)\hat{\phi}_n(x,t)$ |

## 4.3 Mass transport as a banded linear Ordinary Differential Equation (ODE)

To efficiently solve the mass transport equation, we reformulate it as a banded linear ODE. Specifically, we set the upper bound for $x$ to be $M$, and set $\tilde{\phi}(t)$ to be a vector that includes the value of $\tilde{\phi}(x,t)$ for each integer $x$: $\tilde{\phi}(t) = (\tilde{\phi}(0,t), \tilde{\phi}(1,t), \cdots, \tilde{\phi}(M,t))^\top$. With this representation of the conditional mass, the mass transport equation in (4) can be expressed as a simple banded linear ODE:

$$\tilde{\phi}(t)' = \boldsymbol{Q}(t)\tilde{\phi}(t), \tag{5}$$

where $\tilde{\phi}(t)' = (\tilde{\phi}_t(0,t), \cdots, \tilde{\phi}_t(M,t))^\top$, and the matrix $\boldsymbol{Q}(t)$ is a sparse bi-diagonal matrix with $Q_{i,i} = -\lambda(t)$ and $Q_{i-1,i} = \lambda(t)$. The following equation visualizes the ODE in (5) when $M = 2$.

$$\begin{pmatrix} \tilde{\phi}_t(0,t) \\ \tilde{\phi}_t(1,t) \\ \tilde{\phi}_t(2,t) \end{pmatrix} = \begin{pmatrix} -\lambda(t) & & \\ \lambda(t) & -\lambda(t) & \\ & \lambda(t) & -\lambda(t) \end{pmatrix} \begin{pmatrix} \tilde{\phi}(0,t) \\ \tilde{\phi}(1,t) \\ \tilde{\phi}(2,t) \end{pmatrix}. \tag{6}$$

This dynamic ODE is a compact representation of the transport equation in (4) and $M$ decides the dimension of the ODE in (5). In theory, $M$ can be unbounded. However, the conditional probability mass is tends to zero when $M$ becomes large. Hence, in practice we choose a finite support $\{0, 1, \cdots, M\}$ for the conditional probability mass function. To choose a proper $M$, we generate samples from the point process. Suppose the largest number of events in the samples is $L$, we set $M = 2L$ such that it is reasonably large. Next, with the initial probability mass $\tilde{\phi}(t_0) = (1, 0, \cdots, 0)^\top$, we present an efficient algorithm to solve the ODE.

## 4.4 Scalable algorithm for solving the ODE

We present the algorithm that transports the initial mass $\tilde{\phi}(t_0)$ to $\tilde{\phi}(t)$ by solving the ODE.

Since the intensity function is history-dependent and has a discrete jump when an event happens at time $t_k$, the matrix $\boldsymbol{Q}(t)$ in the ODE is discontinuous at $t_k$. Hence we split $[0, t]$ into intervals $[t_k, t_{k+1}]$. On each interval, the intensity is continuous and we can use the classic numerical Runge-Kutta (RK) method [7] to solve the ODE. Figure 2 illustrates the overall algorithm.

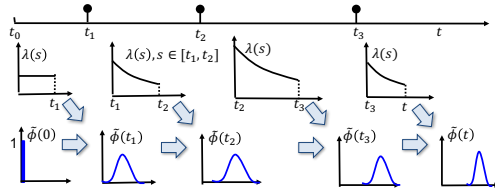

Figure 2: Illustration of Algorithm 1 using Hawkes process. The intensity is updated after each event $t_k$. Within $[t_k, t_{k+1}]$, we use $\phi(t_k)$ and the intensity $\lambda(s)$ to solve the ODE and obtain $\phi(t_{k+1})$.

Our algorithm works as follows. First, with the initial intensity on $[0, t_1]$ and $\tilde{\phi}(t_0)$ as input, the RK method solves the ODE on $[0, t_1]$ and outputs $\tilde{\phi}(t_1)$. Since an event happens at $t_1$, the intensity is updated on $[t_1, t_2]$. Next, with the updated intensity and $\tilde{\phi}(t_1)$ as the initial value, the RK method solves the ODE on $[t_1, t_2]$ and outputs $\tilde{\phi}(t_2)$. This procedure repeats for each $[t_k, t_{k+1}]$ until time $t$.

Now we present the RK method that solves the ODE on each interval $[t_k, t_{k+1}]$. RK divides this interval into equally-spaced subintervals $[\tau_i, \tau_{i+1}]$, for $i = 0, \cdots, I$ and $\Delta\tau = \tau_{i+1} - \tau_i$. It then conducts linear extrapolation on each subinterval. It starts from $\tau_0 = t_k$ and uses $\tilde{\phi}(\tau_0)$ and the approximation of the gradient $\tilde{\phi}(\tau_0)'$ to compute $\tilde{\phi}(\tau_1)$. Next, $\tilde{\phi}(\tau_1)$ is taken as the initial value and the process is repeated until $\tau_I = t_{k+1}$. Appendix D contains details of this method.

The RK method approximates the gradient $\tilde{\phi}(t)'$ with different levels of accuracy, called states $s$. When $s = 1$, it is the Euler method, which uses the first order approximation $\tilde{\phi}(\tau_{i+1}) - \tilde{\phi}(\tau_i)/\Delta\tau$.

We use the ODE45 solver in MATLAB and choose the stage $s = 4$ for RK. Moreover, the main computation in the RK method comes from the matrix-vector product. Since the matrix $\boldsymbol{Q}(t)$ is sparse and bi-diagonal with $O(M)$ non-zero elements, the cost for this operation is only $O(M)$.

## 4.5 Hybrid inference machine with mass transport equation

With the conditional probability mass, we are now ready to express $g(\mathcal{H}_{t-})$ in closed form and estimate $\mathbb{E}_{\mathcal{H}_{t-}}[g(\mathcal{H}_{t-})]$ using the MC sampling method. We present our framework HYBRID:

*(i)* Generate $n$ samples $\{\mathcal{H}_{t-}^i\}$ from a point process $N(t)$ with a stochastic intensity $\lambda(t)$.
*(ii)* For each sample $\mathcal{H}_{t-}^i$, we compute the value of intensity function $\lambda(s|\mathcal{H}_{s-}^i)$, for each $s \in [0,t]$; then we solve (5) to compute the conditional probability mass $\tilde{\phi}^i(x,t)$.
*(iii)* We obtain the estimator of the probability mass function $\phi(x,t)$ and $\mu(t)$ by taking the average: $\hat{\phi}_n(x,t) = \frac{1}{n}\sum_{i=1}^n \tilde{\phi}^i(x,t), \quad \hat{\mu}_n(t) = \sum_x f(x)\hat{\phi}_n(x,t)$

Algorithm 2 summarizes the above procedure. Next, we discuss two properties of HYBRID.

First, our framework efficiently uses all event information in each sample. In fact, each event $t_k$ influences the transport rate of the conditional probability mass (Figure 2). This feature is in sharp contrast to MC that only uses the information of the total number of events and neglects the differences in event times. For instance, the two samples in Figure 1(a) both have three events and MC treats them equally; hence its estimator is an indicator function $\hat{\phi}_n^{mc}(x,t) = \mathbb{I}[x = 3]$. However, for HYBRID, these samples have different event information and conditional probability mass functions, and our estimator in Figure 1(d) is much more informative than an indicator function.

Moreover, our estimator for the probability mass is unbiased if we can solve the mass transport equation in (4) exactly. To prove this property, we show that the following equality holds for an arbitrary function $f$: $(f, \phi) = \mathbb{E}[f(N(t))] = \mathbb{E}_{\mathcal{H}_{t-}}[g(\mathcal{H}_{t-})] = (f, \mathbb{E}_{\mathcal{H}_{t-}}[\tilde{\phi}])$. Then $\mathbb{E}_{\mathcal{H}_{t-}}[\hat{\phi}_n] = \phi$ follows from the fundamental lemma of the calculus of variations [14]. Appendix B contains detailed derivations. In practice, we choose a reasonable finite support for the conditional probability mass in order to solve the mass transport ODE in (5). Hence our estimator is nearly unbiased.

# 5 Applications and extensions to multi-dimensional point processes

In this section, we present two real world applications, where the point process models have intertwined stochasticity and co-evolving intensity functions.

**Predicting the activeness and popularity of users in social networks**. The co-evolution model [12] uses a Hawkes process $N_{us}(t)$ to model information diffusion (tweets/retweets), and a survival process $A_{us}(t)$ to model the dynamics of network topology (link creation process). The intensity of $N_{us}(t)$ depends on the network topology $A_{us}(t)$, and the intensity of $A_{us}(t)$ also depends on $N_{us}(t)$; hence these processes co-evolve over time. We focus on two tasks in this model: (i) inferring the activeness of a user by $\mathbb{E}[\sum_u N_{us}(t)]$, which is the number of tweets and retweets from user $s$; and (ii) inferring the popularity of a user by $\mathbb{E}[\sum_u A_{us}(t)]$, which is the number of new links created to the user.

**Predicting the popularity of items in recommender systems**. Recent works on recommendation systems [10, 25] use a point process $N_{ui}(t)$ to model user $u$'s sequential interaction with item $i$. The intensity function $\lambda_{ui}(t)$ denotes user's interest to the item. As users interact with items over time, the user latent feature $\boldsymbol{u}_u(t)$ and item latent feature $\boldsymbol{i}_u(t)$ co-evolve over time, and are mutually dependent [25]. The intensity is parameterized as $\lambda_{ui}(t) = \eta_{ui} + \boldsymbol{u}_u(t)^\top \boldsymbol{i}_i(t)$, where $\eta_{ui}$ is a baseline term representing the long-term preference, and the tendency for $u$ to interact with $i$ depends on the compatibility of their instantaneous latent features $\boldsymbol{u}_u(t)^\top \boldsymbol{i}_i(t)$. With this model, we can infer an item's popularity by evaluating $\mathbb{E}[\sum_u N_{ui}(t)]$, which is the number of events happened to item $i$.

To solve these prediction tasks, we extend the transport equation to the multivariate case. Specifically, we create a new stochastic process $x(t) = \sum_u N_{us}(t)$ and compute its conditional mass function.

**Theorem 3** (Mass Transport for Multidimensional Point Processes)**.** *Let $N_{us}(t)$ be the point process with intensity $\lambda_{us}(t)$, $x(t) = \sum_{u=1}^U N_{us}(t)$, and $\tilde{\phi}(x,t) = \mathbb{P}[x(t) = x|\mathcal{H}_{t-}]$ be the conditional probability mass of $x(t)$; then $\tilde{\phi}$ satisfies: $\tilde{\phi}_t = -\left(\sum_u \lambda_{us}(t)\right)\tilde{\phi}(x,t) + \left(\sum_u \lambda_{us}(t)\right)\tilde{\phi}(x-1,t)$.*

To compute the conditional probability mass, we also solve the ODE in (5), where the diagonal and off-diagonal of $\boldsymbol{Q}(t)$ is now the negative and positive summation of intensities in all dimensions.

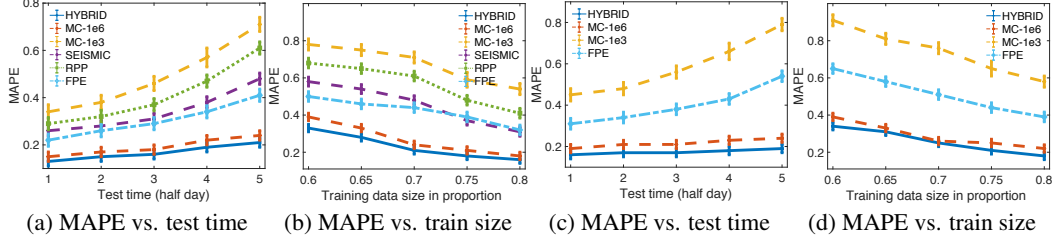

(a) MAPE vs. test time    (b) MAPE vs. train size    (c) MAPE vs. test time    (d) MAPE vs. train size

Figure 3: Prediction results for user activeness and user popularity. (a,b) user activeness: predicting the number of posts per user; (c,d) user popularity: predicting the number of new links per user. Test times are the relative times after the end of train time. The train data is fixed with 70% of total data.

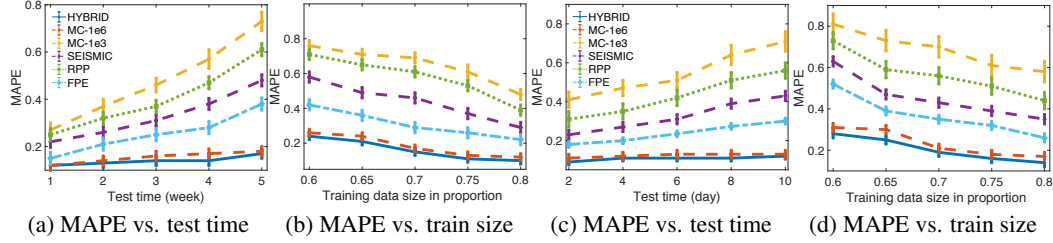

(a) MAPE vs. test time    (b) MAPE vs. train size    (c) MAPE vs. test time    (d) MAPE vs. train size

Figure 4: Prediction results for item popularity. (a,b) predicting the number of watching events per program on IPTV; (c,d) predicting the number of discussions per group on Reddit.

## 6 Experiments

In this section, we evaluate the predictive performance of HYBRID in two real world applications in Section 5 and a synthetic dataset. We use the following metrics:

(i) Mean Average Percentage Error (MAPE). Given a prediction time $t$, we compute the MAPE $|\hat{\mu}_n(t) - \mu(t)|/\mu(t)$ between the estimated value and the ground truth.

(ii) Rank correlation. For all users/items, we obtain two lists of ranks according to the true and estimated value of user activeness/user popularity/item popularity. The accuracy is evaluated by the Kendall-$\tau$ rank correlation [18] between two lists.

### 6.1 Experiments on real world data

We show HYBRID has both accuracy and efficiency improvement in predicting the activeness and popularity of users in social networks and predicting the popularity of items in recommender systems.

**Competitors**. We use $10^3$ samples for HYBRID and compare it with the following the state of the art.

- SEISMIC [32]. It defines a self-exciting process with a post infectiousness factor. It uses the branching property of Hawkes process and heuristic corrections for prediction.
- RPP [13]. It adds a reinforcement coefficient to Poisson process that depicts the self-excitation phenomena. It sets $dN(t) = \lambda(t)dt$ and solves a deterministic equation for prediction.
- FPE [29]. It uses a deterministic function to approximate the stochastic intensity function.
- MC-1E3. It is the MC sampling method with $10^3$ samples (same as these for HYBRID), and MC-1E6 uses $10^6$ samples.

#### 6.1.1 Predicting the activeness and popularity of users in social networks

We use a Twitter dataset [2] that contains 280,000 users with 550,000 tweet, retweet, and link creation events during Sep. 21 - 30, 2012. This data is previously used to validate the network co-evolution model [12]. The parameters for tweeting/retweeting processes and link creation process are learned using maximum likelihood estimation [12]. SEISMIC and RPP are not designed for the popularity prediction task since they do not consider the evolution of network topology. We use $p$ proportion of total data as the training data to learn parameters of all methods, and the rest as test data. We make predictions for each user and report the averaged results.

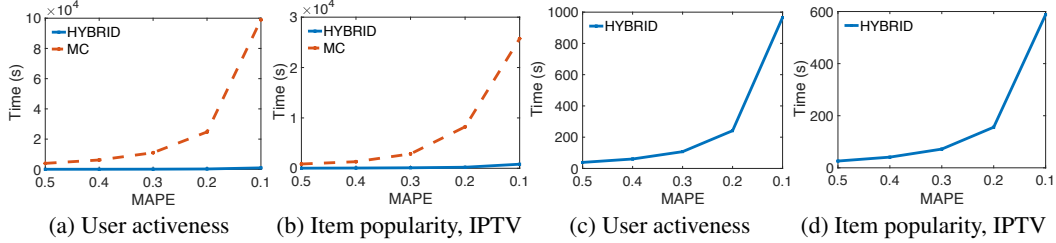

| (a) User activeness | (b) Item popularity, IPTV | (c) User activeness | (d) Item popularity, IPTV |

Figure 5: Scalability analysis: computation time as a function of error. (a,b) comparison between HYBRID and MC in different problems; (c,d) scalability plots for HYBRID.

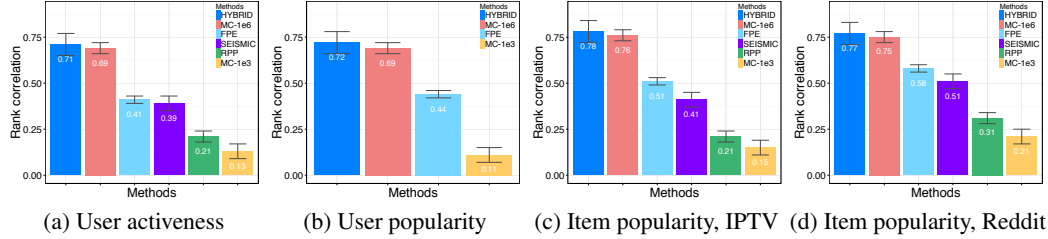

| (a) User activeness | (b) User popularity | (c) Item popularity, IPTV | (d) Item popularity, Reddit |

Figure 6: Rank correlation results in different problems. We vary the proportion $p$ of training data from 0.6 to 0.8, and the error bar represents the variance over different training sets.

**Predictive performance**. Figure 3(a) shows that MAPE increases as test time increases, since the model's stochasticity increases. HYBRID has the smallest error. Figure 3(b) shows that MAPE decreases as training data increases since model parameters are more accurate. Moreover, HYBRID is more accurate than SEISMIC and FPE with only 60% of training data, while these works need 80%. Thus, we make accurate predictions by observing users in the early stage. This feature is important for network moderators to identify malicious users and suppress the propagation undesired content.

Moreover, the consistent performance improvement shows two messages: (i) *considering all the randomness is important*. HYBRID is $2\times$ more accurate than SEISMIC and FPE because HYBRID naturally considers all the stochasticity, but SEISMIC, FPE, and RPP need heuristics or approximations that discard parts of the stochasticity; (ii) *sampling efficiently is important*. To consider all the stochasticity, we need to use the sampling scheme, and HYBRID has a much smaller sample size. Specifically, HYBRID uses the same $10^3$ samples, but has $4\times$ error reduction compared with MC-1E3. MC-1E6 has a similar predictive performance as HYBRID, but needs $10^3\times$ more samples.

**Scalability**. How does the reduction in sample size improve the speed? Figure 5(a) shows that as the error decreases from 0.5 to 0.1, MC has higher computation cost, since it needs much more samples than HYBRID to achieve the same error. We include the plots of HYBRID in (c). In particular, to achieve the error of 0.1, MC needs $10^6$ samples in 27.8 hours, but HYBRID only needs 14.4 minutes with $10^3$ samples. We use the machine with 16 cores, 2.4 GHz Intel Core i5 CPU and 64 GB memory.

**Rank correlation**. We rank all users according to the predicted level of activeness and level of popularity separately. Figure 6(a,b) show that HYBRID performs the best with the accuracy around 80%, and it consistently identifies around 30% items more correctly than FPE on both tasks.

### 6.1.2 Predicting the popularity of items in recommender systems

In the recommendation system setting, we use two datasets from [25]. The IPTV dataset contains 7,100 users' watching history of 436 TV programs in 11 months, with around 2M events. The Reddit dataset contains online discussions of 1,000 users in 1,403 groups, with 10,000 discussion events. The predictive and scalability performance are consistent with the application in social networks. Figure 4 shows that HYBRID is 15% more accurate than FPE and 20% than SEISMIC. Figure 5 also shows that HYBRID needs much smaller amount of computation time than MC-1E6. To achieve the error of 0.1, it takes 9.8 minutes for HYBRID and 7.5 hours for MC-1E6. Figure 6(c,d) show that HYBRID achieves the rank correlation accuracy of 77%, with 20% improvement over FPE.

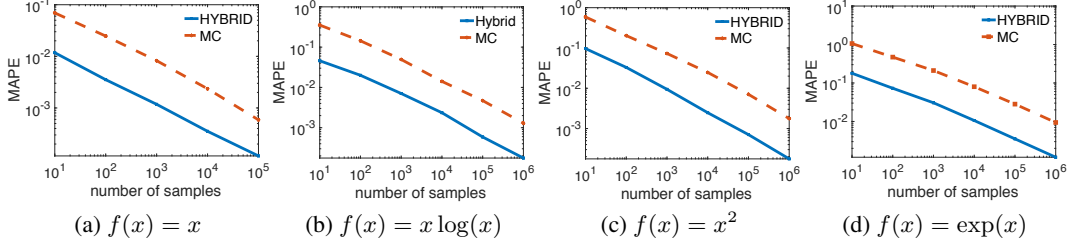

(a) $f(x) = x$      (b) $f(x) = x\log(x)$      (c) $f(x) = x^2$      (d) $f(x) = \exp(x)$

Figure 7: Error of $\mathbb{E}[f(N(t))]$ as a function of sample size (loglog scale). (a-d) different choices of $f$.

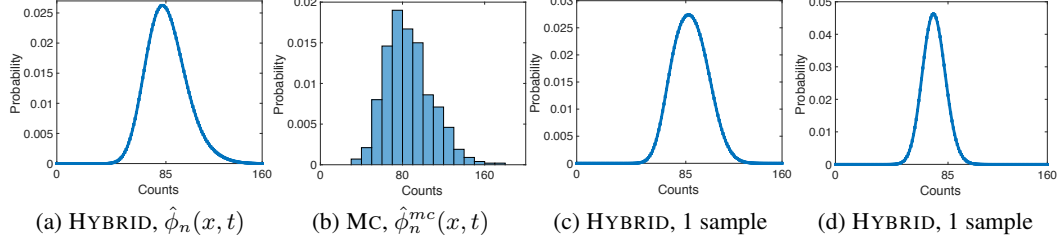

(a) HYBRID, $\hat{\phi}_n(x, t)$    (b) MC, $\hat{\phi}_n^{mc}(x, t)$    (c) HYBRID, 1 sample    (d) HYBRID, 1 sample

Figure 8: Comparison of estimators of probability mass functions in HYBRID and MC. (a,b) estimators with the same 1000 samples. (c,d) estimator with one sample in HYBRID.

## 6.2 Experiments on synthetic data

We compare HYBRID with MC in two aspects: (i) the significance of the reduction in the error and sample size, and (ii) estimators of the probability mass function. We study a Hawkes process and set the parameters of its intensity function as $\eta = 1.2$, and $\alpha = 0.5$. We fix the prediction time to be $t = 30$. The ground truth is computed with $10^8$ samples from MC simulations.

**Error vs. number of samples**. In four tasks with different $f$, Figure 7 shows that given the same number of samples, HYBRID has a smaller error. Moreover, to achieve the same error, HYBRID needs $100\times$ less samples than MC. In particular, to achieve the error of $0.01$, (a) shows HYBRID needs $10^3$ and MC needs $10^5$ samples; (b) shows HYBRID needs $10^4$ and MC needs $10^6$ samples.

**Probability mass functions**. We compare our estimator of the probability mass with MC. Figure 8(a,b) show that our estimator is much smoother than MC, because our estimator is the average of conditional probability mass functions, which are computed by solving the mass transport equation. Moreover, our estimator centers around 85, which is the ground truth of $\mathbb{E}[N(t)]$, while that of MC centers around 80. Hence HYBRID is more accurate. We also plot two conditional mass functions in (c,d). The average of 1000 conditional mass functions yields (a). Thus, this averaging procedure in HYBRID adjusts the shape of the estimated probability mass. On the contrary, given one sample, the estimator in MC is just an indicator function and cannot capture the shape of the probability mass.

## 7 Conclusions

We have proposed HYBRID, a generic framework with a new formulation of the prediction problem in point processes and a novel mass transport equation. This equation efficiently uses the event information to update the transport rate and compute the conditional mass function. Moreover, HYBRID is applicable to general point processes and prediction tasks with an arbitrary function $f$. Hence it can take any point process models as input, and the predictive performance of our framework can be further improved with the advancement of point process models. Experiments on real world and synthetic data demonstrate that HYBRID outperforms the state of the art both in terms of accuracy and efficiency. There are many interesting lines for future research. For example, HYBRID can be generalized to marked point processes [4], where a mark is observed along with the timing of each event.

**Acknowledgements**. This project was supported in part by NSF IIS-1218749, NIH BIGDATA 1R01GM108341, NSF CAREER IIS-1350983, NSF IIS-1639792 EAGER, NSF CNS-1704701, ONR N00014-15-1-2340, DMS-1620342, CMMI-1745382, IIS-1639792, IIS-1717916, NVIDIA, Intel ISTC and Amazon AWS.

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
