[Supplementary Material]

# A  Proof of Theorem 2

**Theorem 2** (Mass Transport Equation for Point Processes). *Let $\lambda(t) := \lambda(t|\mathcal{H}_{t-})$ be the conditional intensity function of the point process $N(t)$ and $\tilde{\phi}(x,t) := \mathbb{P}[N(t) = x|\mathcal{H}_{t-}]$ be its conditional probability mass function; then $\tilde{\phi}(x,t)$ satisfies the following differential-difference equation:*

$$\tilde{\phi}_t(x,t) := \frac{\partial\tilde{\phi}(x,t)}{\partial t} = \begin{cases} -\lambda(t)\tilde{\phi}(x,t) + \lambda(t)\tilde{\phi}(x-1,t) & \text{if } x = 1,2,3,\cdots \\ -\lambda(t)\tilde{\phi}(x,t) & \text{if } x = 0 \end{cases} \tag{7}$$

*Proof.* For the simplicity of notation, we define a functional operator $\mathcal{F}[\tilde{\phi}]$ as follows:

$$\mathcal{F}[\tilde{\phi}] = -\lambda(t)\tilde{\phi}(x,t) + \lambda(t)\tilde{\phi}(x-1,t)\mathbb{I}[x \geqslant 1],$$

where $\mathbb{I}(\cdot)$ is an indicator function.

Our goal is to prove $\mathcal{F}[\tilde{\phi}] = \tilde{\phi}_t$. For the simplicity of notation, we define the inner product [11] between functions $f(x)$ and $g(x)$ as the summation of the product of of $f(x)$ and $g(x)$, where $x \in \mathbb{N}$:

$$(f,g) = \sum_{x=0}^{\infty} f(x)g(x)$$

To prove the equality $\tilde{\phi}_t = \mathcal{F}[\tilde{\phi}]$, we will prove that the equality $(v, \tilde{\phi}_t) = (v, \mathcal{F}[\tilde{\phi}])$ holds for any test function $v(x)$. Then the equality $\tilde{\phi}_t = \mathcal{F}[\tilde{\phi}]$ follows from the famous Fundamental Lemma of Calculus of Variations [15]. To show the above equality, we start by computing $(v, \tilde{\phi}_t)$.

**Computing** $(v, \tilde{\phi}_t)$. According to the definition of expectation and the fact that $\tilde{\phi}(x,t)$ is the conditional probability mass, we have

$$\mathbb{E}[v(N(t))|\mathcal{H}_{t-}] = \sum_{x=0}^{\infty} v(x)\mathbb{P}[N(t) = x|\mathcal{H}_{t-}] = \sum_{x=0}^{\infty} v(x)\tilde{\phi}(x,t) = (v, \tilde{\phi}).$$

Taking the gradient with respect to $t$ yields

$$\frac{\partial\mathbb{E}[v(N(t))|\mathcal{H}_{t-}]}{\partial t} = \sum_{x=0}^{\infty} v(x)\tilde{\phi}_t(x,t) = (v, \tilde{\phi}_t). \tag{8}$$

Next, we obtain another expression for $(v, \tilde{\phi}_t)$. First we show the following property of $\mathrm{d}v(N(t))$

$$\mathrm{d}v(N(t)) = \big(v(N(t)+1) - v(N(t))\big)\mathrm{d}N(t) \tag{9}$$

In fact, from the definition of the differential operator $\mathrm{d}$, we have the following property:

$$\mathrm{d}v(N(t)) := v\big(N(t+\mathrm{d}t)\big) - v\big(N(t)\big) = v\big(N(t)+\mathrm{d}N(t)\big) - v\big(N(t)\big)$$

Since $\mathrm{d}N(t) = \{0,1\}$, if $\mathrm{d}N(t) = 0$, we have $\mathrm{d}v(N(t)) = 0$; otherwise, we have $\mathrm{d}v(N(t)) = v(N(t)+1) - v(N(t))$. For both cases, equation (9) holds.

Next, we integrate both sides of (9) on $[0,t]$ and express $v(N(t))$ as follows:

$$v(N(t)) = v(N(0)) + \int_0^t \big(v(N(t)+1) - v(N(t))\big)\mathrm{d}N(t) \tag{10}$$

Given $\mathcal{H}_{t-}$, we take the conditional expectation of (10) and obtain the following expression:

$$\mathbb{E}[v(N(t))|\mathcal{H}_{t-}] = v(N(0)) + \mathbb{E}\Big[\int_0^t \big(v(N(t)+1)\big) - v(N(t))\big)\lambda(t)\mathrm{d}t\Big|\mathcal{H}_{t-}\Big] \tag{11}$$

Now we differentiate both sides of (11) with respect to time $t$ and obtain the following expression:

$$\begin{aligned} \frac{\partial\mathbb{E}[v(N(t))|\mathcal{H}_{t-}]}{\partial t} &= \mathbb{E}\Big[\frac{\partial}{\partial t}\int_{t_0}^t \big(\mathcal{B}[v](N(s))\big)\mathrm{d}s\Big|\mathcal{H}(t^-)\Big] \\ &= \mathbb{E}\Big[\mathcal{B}[v](N(t))\Big|\mathcal{H}_{t-}\Big] \\ &= \sum_{x=0}^{\infty} \mathcal{B}[v](x(t))\tilde{\phi}(x,t) \\ &= (\mathcal{B}[v], \tilde{\phi}) \end{aligned} \tag{12}$$

where $\mathcal{B}[v]$ is another functional operator defines as

$$\mathcal{B}[v]\big(N(t)\big) = \big(v(N(t)+1) - v(N(t))\big)\lambda(t) \tag{13}$$

Since (12) and (8) are equivalent, we have:

$$(v, \tilde{\phi}_t) = (\mathcal{B}[v], \tilde{\phi})$$

Now we have finished the first part of the proof. In the second part, our goal is to move the operator $\mathcal{B}$ from test function $v$ to the conditional probability mass function $\phi$ and prove $(\mathcal{B}[v], \tilde{\phi}) = (v, \mathcal{F}[\tilde{\phi}])$. We start by computing $(\mathcal{B}[v], \tilde{\phi})$ as follows.

**Computing** $(\mathcal{B}[v], \tilde{\phi})$. We define a new post-jump variable as $y = x + 1$, and conduct a *change of variable* from $x$ to $y = x + 1$ in $(\mathcal{B}[v], \tilde{\phi})$. Specifically, we express $(\mathcal{B}[v], \tilde{\phi})$ as follows

$$\sum_{x=0}^{\infty} \big(v(x+1) - v(x)\big)\lambda(t)\tilde{\phi}(x,t) = \sum_{x=0}^{\infty} v(x+1)\lambda(t)\tilde{\phi}(x,t) - \sum_{x=0}^{\infty} v(x)\lambda(t)\tilde{\phi}(x,t)$$

$$= \sum_{y=1}^{\infty} v(y)\lambda(t)\tilde{\phi}(y-1,t) - \sum_{x=0}^{\infty} v(x)\lambda(t)\tilde{\phi}(x,t) \tag{14}$$

Next, we use an indicator function and let the value of $y$ to start from $0$ in the first term of (14):

$$\sum_{y=1}^{\infty} v(y)\lambda(t)\tilde{\phi}(y-1,t) = \sum_{y=0}^{\infty} v(y)\lambda(t)\tilde{\phi}(y-1,t)\mathbb{I}[y \geqslant 1]$$

$$= \Big(v(y), \lambda(t))\tilde{\phi}(y-1,t)\mathbb{I}[y \geqslant 1]\Big) \tag{15}$$

Now we substitute (15) back to (14) and obtain the following equation:

$$\sum_{x=0}^{\infty} \big(v(x+1) - v(x)\big)\lambda(t)\tilde{\phi}(x,t) = \Big(v(y), \lambda(t))\tilde{\phi}(y-1,t)\mathbb{I}[y \geqslant 1]\Big) - \Big(v(x), \lambda(t)\tilde{\phi}(x,t)\Big)$$

$$= \Big(v(x), \lambda(t))\tilde{\phi}(x-1,t)\mathbb{I}[x \geqslant 1]\Big) - \Big(v(x), \lambda(t)\tilde{\phi}(x,t)\Big)$$

$$= (v, \mathcal{F}[\tilde{\phi}]) \tag{16}$$

Hence, for an arbitrary function $v(x)$, we have shown the following equality:

$$(v, \tilde{\phi}_t) = (\mathcal{B}[v], \tilde{\phi}) = (v, \mathcal{F}[\tilde{\phi}]).$$

This yields $\tilde{\phi}_t = \mathcal{F}[\tilde{\phi}]$ and the proof is now complete. $\qquad\square$

# B  Proof of unbiasedness of the estimator for the probability mass function

We just need to show the following equality: $\phi(x,t) = \mathbb{E}_{\mathcal{H}_{t^-}}[\tilde{\phi}(x,t)]$. For the simplicity of notation, we define the inner product between functions $f(x)$ and $g(x)$ as $(f,g) := \sum_x f(x)g(x)$, where $x \in \mathbb{N}$.

First, according to the definition of expectation, we have

$$\mathbb{E}[f(N(t))] := (f, \phi)$$

Next, from the definition of conditional probability mass, $g(\mathcal{H}_{t^-})$ can be expressed as

$$g(\mathcal{H}_{t^-}) = \sum_x f(x)\tilde{\phi}(x,t) = (f, \tilde{\phi}) \tag{17}$$

Taking expectation to both sides of (17) yields

$$\mathbb{E}_{\mathcal{H}_{t^-}}[g(\mathcal{H}_{t^-})] = (f, \mathbb{E}_{\mathcal{H}_{t^-}}[\tilde{\phi}])$$

Finally, since $\mathbb{E}[f(N(t))] = \mathbb{E}_{\mathcal{H}_{t^-}}[g(\mathcal{H}_{t^-})]$, we have $(f, \tilde{\phi}) = (f, \mathbb{E}_{\mathcal{H}_{t^-}}[\tilde{\phi}])$, which holds for an arbitrary function $f$. Hence the equality $\mathbb{E}_{\mathcal{H}_{t^-}}[\tilde{\phi}] = \phi$ follows from the Fundamental Lemma of Calculus of Variations [15].

## C   Proof of Theorem 1

**Theorem 1.** *For time $t > 0$ and an arbitrary function $f$, we have:*

$$\mathbb{VAR}[g(\mathcal{H}_{t-})] < \mathbb{VAR}[f(N(t))] \tag{18}$$

*Proof.* The proof contains two steps. We first compute the expected value of the conditional variance $\mathbb{E}\left[\mathbb{VAR}\left[f(N(t))|\mathcal{H}_{t-}\right]\right]$, and next compute the variance of the conditional expected value $\mathbb{VAR}\left[g(\mathcal{H}_{t-})\right]$.

*(i) Expected value of the conditional variance.* Since $\mathbb{VAR}[f(N(t))|\mathcal{H}_{t-}]$ is a random variable, we can compute its expected value. Using the definition of variance, *i.e.*, $\mathbb{VAR}[f(N(t))|\mathcal{H}_{t-}] = \mathbb{E}[f(N(t))^2|\mathcal{H}_{t-}] - [\mathbb{E}[f(N(t))|\mathcal{H}_{t-}]]^2$, we have

$$\mathbb{E}\left[\mathbb{VAR}\left[f(N(t))|\mathcal{H}_{t-}\right]\right] = \mathbb{E}\left[\mathbb{E}\left[f(N(t))^2|\mathcal{H}_{t-}\right]\right] - \mathbb{E}\left[\left[f(N(t))|\mathcal{H}_{t-}\right]^2\right] \tag{19}$$

$$= \mathbb{E}[f(N(t))^2] - \mathbb{E}\left[\left[\mathbb{E}[f(N(t))|\mathcal{H}_{t-}]\right]^2\right] \tag{20}$$

*(ii) Variance of the conditional expected value.* We express $\mathbb{VAR}\left[g(\mathcal{H}_{t-})\right]$ as follows

$$\mathbb{VAR}\left[g(\mathcal{H}_{t-})\right] = \mathbb{VAR}\left[\mathbb{E}\left[f(N(t))|\mathcal{H}(t)\right]\right] \tag{21}$$

$$= \mathbb{E}\left[\mathbb{E}\left[f(N(t))|\mathcal{H}_{t-}\right]^2\right] - \left[\mathbb{E}\left[\mathbb{E}[f(N(t))|\mathcal{H}_{t-}]\right]\right]^2 \tag{22}$$

$$= \mathbb{E}\left[\mathbb{E}\left[f(N(t))|\mathcal{H}_{t-}\right]^2\right] - \mathbb{E}[f(N(t))]^2 \tag{23}$$

Combining (20) and (23) yields the following equation:

$$\mathbb{VAR}[g(\mathcal{H}_{t-})] + \mathbb{E}\left[\mathbb{VAR}\left[f(N(t))|\mathcal{H}_{t-}\right]\right] = \mathbb{VAR}[N(t)]$$

Next, we show that the inequality in our theorem is strict. According to the definition of counting process, we have $N(0) = 0$. Moreover, we are only interested in the scenarios where the number of events are positive, *i.e.*, $N(t) > 0$ for future time $t > 0$. Since the point process $N(t)$ is right continuous and not a predictable process [4], we obtain the fact that conditioning on $\mathcal{H}_{t-}$, there is a stochastic jump at time $t$ and the value of $f(N(t))$ is random and not a constant. Hence the conditional variance $\mathbb{VAR}\left[f(N(t))|\mathcal{H}_{t-}\right]$ is positive and we have $\mathbb{E}\left[\mathbb{VAR}\left[f(N(t))|\mathcal{H}_{t-}\right]\right] > 0$. The proof is now complete.

□

# D    Details on the Runge-Kutta (RK) method

We present details of the RK method. For the simplicity of notation, we set $\tilde{\phi}'(t) = f(\tilde{\phi}, t) = Q(t)\tilde{\phi}(t)$.

The RK method divides the interval $[t_k, t_{k+1}]$ into intervals $[\tau_i, \tau_{i+1}]$, for $i = 0, \cdots, I$, with $\Delta\tau = \tau_{i+1} - \tau_i$. This method conducts linear extrapolation on contiguous subintervals $[\tau_i, \tau_{i+1}]$. Specifically, it starts from $\tau_0 := t_k$, and within $[\tau_0, \tau_1]$ the RK method of *stage s* computes $y_m = f(\tilde{\phi}_m, \tau_0 + \Delta\tau c_m)$ at $s$ recursively defined input locations, for $m = 1, \cdots, s$, where $\tilde{\phi}_m$ is computed as a linear combination of previous $y_{n<m}$ as $\tilde{\phi}_m = \tilde{\phi}_0 + \Delta\tau \sum_{n=1}^{m-1} w_{mn} y_n$. Then, it returns the prediction for the solution at $\tau_1$ as $\tilde{\phi}(\tau_0 + \Delta\tau)$. In the compact form,

$$y_m = f\left(\tilde{\phi}_0 + \Delta\tau \sum_{n=1}^{m-1} w_{mn} y_n, \tau_0 + \Delta\tau c_m\right), \ m = 1, \cdots, s, \ \tilde{\phi}(\tau_0 + \Delta\tau) = \tilde{\phi}_0 + \Delta\tau \sum_{m=1}^{s} b_m y_m$$

Next, $\tilde{\phi}(\tau_0 + \Delta\tau)$ is taken as the initial value for $\tau_1 = \tau_0 + \Delta\tau$ and the process is repeated until $\tau_I := t_{k+1}$. Note that RK outputs the conditional probability mass at all timestamps $\{\tau_i\}$; hence it captures the mass transport on $[t_k, t_{k+1}]$.

The main computation in RK is the matrix-vector product. Since the matrix $Q(t)$ is sparse and bi-diagonal with $O(M)$ non-zero elements, the cost for this operation is only $O(M)$.