[Reviews · NeurIPS 2017]

Reviewer 1



In my opinion this paper is quite good, although I do not consider myself an expert on point processes, which should be taken into account when reading this review. I consider it in the top 50% of NIPS papers as a lower bound, admitting that I do not know enough to evaluated it further. I have asked the committee to ensure that at least one person who can judge the proofs sufficiently be assigned to this paper and they promised me this would be the case. I do hope if is published as I would like to try using it. Overall I think the main issue with this paper is that it would more comfortably fit in about 20 pages. Here are my detailed comments within the scope of my expertise: Coming from a background of modeling user activity as someone who interprets time series regarding heart rate and accelerators, this paper was not what I expected based on the title. To some extent I think the paper would have been better titled "HYBRID: a framework that provides an unbiased estimator of the probability mass function of point processes," but I do see why you called it this after reading the paper. There are many contributions to in this paper that comprise the framework: the reformulation of the prediction problem using a conditional expectation that incorporates history, allowing sample efficiency for prediction due to a lower variance in the new estimator, the derivation of the differential-difference equation to compute the conditional estimator and both a theoretical evaluation of the framework on synthetic data as well as two evaluations on real world data. Overall, the paper seems quite good and well supported. It is also well written. Prior Art: My one comment on the prior art is that one cluster of author (the authors of 5,7,8,9,11,12,24,25,26,27 and 31) does seem to dominate the references. These citations are necessary as the authors of this document do need to reference this material, however, for the datasets used it would be more appropriate to cite the original authors who contributed the data rather than simply 12. Demetris Antoniades and Constantine Dovrolis. Co-evolutionary dynamics in social networks: A case study of twitter. Computational Social Networks, 2(1):14, 2015. My guess is that since there is so much material here and 12 does need to be cited as well the authors were trying to save space, this would be a more appropriate citation for the dataset. Similarly, I believe this paper: Yehia Elkhatib ; Mu Mu ; Nicholas Race "Dataset on usage of a live & VoD P2P IPTV service" P2P, London, UK 2014 is a more appropriate citation for one of the datasets data used in Section 6.1.2 cited as "the datasets used in [24] The other dataset "used in [24]" seems to be also the Antoniades and Dovrolis set. It would be better to reference these directly. Also, it would be great to know if this area of research is really restricted primarily to this one group or if other groups are actively pursuing these topics. Again, I feel it is a likely space limitation issue so perhaps this is more a comment for a journal version of the paper. Solution Overview (Section 3) With respect to the claim to generalizability, could the authors please give an example of a point process that is not covered by prior art (e.g not represented by a reinforced Poisson process or a Hawkes Process) as a person not overly familiar with point processes, this would help me understand better the impact of this work. This was done nicely in lines 31 and 31 for why the generalization of the function is important. In general I think this is well presented, your contribution is a more general framework that can preserves the stochasticity of the function and yet requires fewer samples than MC methods. With respect to the claim of a new random variable, g(H_t_minus), the idea of a conditional intensity function, conditioned on the history of the intensity function is not new, it seems to be a known concept based on a few quick web searches. I have to assume that you mean that using it as a random variable for prediction is a new idea. The closest similar work I found to this is the use of a "mark-specific conditional intensity function" in "Marked Temporal Dynamics Modeling based on Recurrent Neural Network" by Yongqing Wang, Shenghua Liu, Huawei Shen, Xueqi Cheng on arXiv: (https://arxiv.org/pdf/1701.03918.pdf). Again unfortunately by the method of google searching as I am not expert in this area. If you think it is similar you can reference it, if not you can ignore this. Again, it seems related from my point of view, but my expertise is lacking in this area. With respect to the proof contained in Appendix C: It looks good to me. I had to look up taking the expectation of a conditional random variable, http://www.baskent.edu.tr/~mudogan/eem611/ConditionalExpectation.pdf and the conditions under which a variance could be negative and I believe that you proved your point well. It can't negative or zero because it right progressing and there is an impulse at t(0) which makes it non-constant and therefore non-zero if I understand you correctly. I am hoping supplemental material will be included in the proceedings. With respect to the contribution of the formulation of the novel transport equation, this does seem novel and useful. I could find nothing similar. The closest match I found was: "Probability approximation of point processes with Papangelou conditional intensity," by Giovanni Luca Torris forthcoming in the Bernoulli sociaty (http://www.bernoulli-society.org/index.php/publications/bernoulli-journal/bernoulli-journal-papers) again based on a web search and not my knowledge, so take it as a paper of potential interest for you not a challenge to your novelty. With respect to the mass transport equation: This does seem like a very good idea. Intrinsically i am understanding that as intensity process generates events these contribute to that mass and that the accumulated mass is modeled as decreasing over time on absence of new events. I understand that the authors used tools from numerical methods for approximating the integration of the probability mass function and in the end solved their equation with ode45 from Matlab. Everything Matlab says about the method is consistent with what is presented here. "ODE23 and ODE45 are functions for the numerical solution of ordinary differential equations. They employ variable step size Runge-Kutta integration methods. ODE23 uses a simple 2nd and 3rd order pair of formulas for medium accuracy and ODE45 uses a 4th and 5th order pair for higher accuracy." I cannot comment with much expertise on the proof in Appendix A. I get the general idea and it seems plausible. I looked up the fundamental lemma of the calculus of variations and tried to follow the math, but I cannot guarantee that I did not miss something as I am not vary familiar with numerical methods. I found Figure 2 very helpful for getting an intuitive understanding for what was being described by the equations. I found the section on event information fairly compelling with respect to understanding why they are able to reduce the samples used for the estimate (increased sample efficiency). For the experiments int Figure 3. Did you train on 70% and test on 30% or did you use a train,test and hold-out set (e.g. 70/15/15)? Overall this looks like quite a promising and solid idea and framework, so I would recommend inclusion at NIPS based on my understanding of the content. Writing: here are some type-os I noticed as I was reading the paper. 10: "state-of-arts" -> "the state of the art" is better 36 "adversely influence" -> "adversely influences" 64: "the phenomena of interests" -> "the phenomena of interest" 75: The root mean square error (RMSE) is defines as -> is defined as

Reviewer 2



This paper provides a framework for analysis of point processes, in particular to predict a function f(N(t)) where N(t) measures the number of times an event has occurred up until a time t. The paper derives an ODE (4) that determines evolution of probability that N takes a particular values at time t (so, the derivative in the ODE is wrt t. After approximating the solution to this ODE, the expectation of f can be taken. This paper is distant from my expertise, so I advise the committee to pay closer attention to other reviews of this paper! From my outsider's perspective, this paper could benefit from an expositional revision at the least. In particular, it's easy to "lose the forest through the trees." The paper gives a nice high-level introduction to the problem and then dives right into math. It'd be great to give a "middle" level of detail (just a paragraph or two) to fill the gap --- e.g. make all the variables lambda, phi, f, N, H_t, ... concrete for a single application---including a clear statement of the input and output data (this wasn't clear to me!). Roughly just a few sentences to the effect "As an example, suppose we wish to predict X from Y. We are given Z (notated using variables W in our paper) and wish to predict A (notated B in our paper)." This would help me contextualize the math quite a bit---I could follow the equations formally but was struggling on an applications side to understand what was going on. Note: * Given the recent popularity of "optimal transport" in machine learning, it might be worth noting that here you refer to "mass transport" as the ODE/PDE for moving around mass rather than anything having to do with e.g. Wasserstein distances. * Theorem 2 is quite natural, and before the proof sketch I'd recommend explaining what's going on in a few words. All this equation is saying is that phi(x,t) changes for two potential reasons: An event occurs in the regime that hasn't reached phi(x,t), bumping up phi(x-1,t) --- this is the second term --- or an extra event occurs and bumps things up to phi(x+1,t) --- the first term. * l.153 --- why should the upper bound for x be known a priori? * l.190(ii) --- is it clear at this point what you mean by "compute the intensity function" --- this seems vague * Paragraph starting l.207 -- what is u?

Reviewer 3



The authors propose Hybrid, a framework to estimate the probability mass function for point processes. They reduce the problem to estimating the mass function conditioned on a given history, and solve the mass transport equation on each intervals to obtain the mass for the future, until the desired time t. This method is shown to have better sampling efficiency compared to MC. I would like the authors to be more clear on the following points: (1) Regarding applying this framework, is the intensity function \lambda(t) known (pre-defined) or learned from the data? I assumed that it is pre-defined since the first step (line 190) of generating history samples depends on the intensity function. (2) When comparing the running time in line 52 and line 269, the criteria is to achieve the same MAPE on \mu. Does the same conclusion still hold when the target is the MAPE of the probability mass function (i.e., P(N(t))) itself?

Reviewer 4



In summary, this work is concerned with variance reduction via Rao-Blackwellization for general point processes. The main insight is to condition on the filtration generated by the process and solve the differential equation describing the time evolution of the corresponding conditional distributions. The latter is facilitated by working on discrete state spaces, made finite with suitable truncation and the use of numerical integration to perform time discretization. Although I find the contribution novel and am convinced of its utility in practical applications, I would like the author/s to be more transparent about the impact of the above-mentioned spatial truncation and temporal discretization. In particular, the claim that their proposed methodology returns unbiased estimators is inaccurate when these approximations are taken into account. That said, I believe debiasing methods described by McLeish (2010) and Rhee and Glynn (2015) may be employed here. Typo on page 5, line 190: the form of the estimator for expectation of test function f is missing averaging over replicates.